# Benzoic Acid Metabolism and Lipopolysaccharide Synthesis of Intestinal Microbiome Affects the Health of Ruminants under Free-Range and Captive Mode

**DOI:** 10.3390/life12071071

**Published:** 2022-07-18

**Authors:** Xuan Fu, Yaopeng Zhang, Bin Shi, Xiaokang Wu, Hongwen Zhao, Zhongbao Xin, Jinshui Yang

**Affiliations:** 1State Key Laboratory of Agrobiotechnology, College of Biological Sciences, China Agricultural University, Beijing 100193, China; sz20213020160@cau.edu.cn (X.F.); yaopengzhang796@gmail.com (Y.Z.); b20193020104@cau.edu.cn (X.W.); hwzhao@cau.edu.cn (H.Z.); 2Key Laboratory of Environmental Nanotechnology and Health Effects Research, Center for Eco-Environmental Sciences, Chinese Academy of Sciences, Beijing 100085, China; binshi@rcees.ac.cn; 3Key Laboratory of State Forestry Administration on Soil and Water Conservation, Beijing Forestry University, Beijing 100083, China; xinbj@bjfu.edu.cn; 4Jixian National Forest Ecosystem Observation and Research Station, Chinese National Ecosystem Research Network, School of Soil and Water Conservation, Beijing Forestry University, Beijing 100083, China

**Keywords:** gut microbes, ruminant, yak, goat, free-range, captive

## Abstract

It is urgent to explore new ways to protect endangered wild animals and develop sustainable animal husbandry on the Qinghai–Tibet Plateau due to its fragile ecological environment. Ruminants, raised in captivity and free-range, have important niches in the Plateau and are the best models to analyze the effects of different feeding modes on their health. In this study, two ruminants, yaks and goats in free-range and captive modes, respectively, were selected to study the relationship between gut microbes and ruminant health. The results showed that the gut microbial diversity of free-range ruminants was higher than those of captive ruminants. Principal co-ordinates analysis (PCoA) showed that there were significant differences in the gut microbial communities in different breeding modes. Both the captive ruminants enriched the Succinivibrionaceae family, which had a strong potential to synthesize lipopolysaccharide, and the low exercise amount of the captive animals was significantly related to this function. Meanwhile, free-range ruminants enriched Oscillospiraceae, which had the potential to degrade benzoic acid, and this potential had a significant positive correlation with resistance to parasitic infections. We offer other possibilities, such as adding benzoic acid to feed or increasing the exercise time of captive ruminants to make them healthier.

## 1. Introduction

The Qinghai–Tibet Plateau, known as the “Third Pole” of the world, has the significant climatic characteristics of low pressure, low oxygen, low temperature, and intense ultraviolet radiation. Its ecological environment is fragile, and its space for human and animal survival and development is limited. In recent years, due to global warming and artificial overgrazing, the ecosystem of the Qinghai–Tibet Plateau has become more sensitive [1]. Yaks are endemic to the Qinghai–Tibet Plateau and have a pivotal ecological position in the Qinghai–Tibet Plateau. They are widely distributed throughout the plateau area of more than 3000 m. Traditional yak breeding is mainly free-range based, but in recent years, with the continuous expansion of yak-breeding scale and decline of pasture production, the breeding method has changed from free-range to house feeding during the cold season and free-range during the warm season, or house feeding throughout the year. Due to the different living environments and feeding patterns, the parasitic infection rates of captive yaks are significantly higher than those of free-range yaks, and they are more prone to various diseases [2].

In addition, endangered wild animals, such as European mouflons (*Ovis orientalis musimon*) and blue sheep (*Pseudois nayaur*) are two kinds of wild goats who are crucial for the ecosystem of the Qinghai–Tibet Plateau. The European mouflon (*Ovis orientalis musimon*), a subspecies of *O. orientalis* and a member of the Bovidae family, is a wild ruminant native to the Mediterranean basin [3], and was listed as a vulnerable species under the A2cde standard by Valdez (2008) [4]. Most European mouflons are kept in zoos around the world. The blue sheep (*Pseudois nayaur*) is a species of Bovidae and is mainly distributed throughout Bhutan, China’s Qinghai–Tibet Plateau and its surrounding mountains, Northern India, and Nepal. Their habitats are generally alpine areas 4000 m above sea level. From 1991 to 2002, the number of blue sheep on the Qinghai–Tibet Plateau sharply declined, and they were listed as a second-class National Protected and endangered species. Currently, it is challenging to observe blue sheep populations in nature. Captive breeding can achieve benign and sustainable development of animal populations under limited space and animal populations, so it is considered an effective method to protect endangered animals [5]. However, like captive yaks, captive European mouflons and blue sheep are more susceptible to digestive tract diseases and parasitic diseases, and are prone to having stress syndromes, such as poor rumination and loss of appetite [6,7]. Therefore, exploring the impact of different farming methods on the health of wild animals has strategic significance for the protection of endangered species.

Gut microbes play important roles in the host’s digestion, nutrient uptake, metabolism, immunity, and infection response [8]. Due to changes in the diets and living environments of captive animals, their gut microbiotas have been greatly altered, a process which affects their overall health, especially their digestive and immune functions [9]. Zhang found that the gut microbial community structure of yaks changed significantly after they were transferred from free-range to house feeding [10]. Tian found that the rumen bacterial diversity and richness of Pengbo semi-fine sheep in the free-range group were significantly higher than those in the house feeding group [11]. Sun found that Firmicutes enriched in wild musk deer can help them to absorb more nutrients from food and produce a large amount of short-chain fatty acids, which can help the musk deer to resist inflammation [12]. Benzoic acid, as an acidifier, was proved to maintain intestinal microecological balance and reduce mortality in piglets [13]. Lipopolysaccharide (LPS) is the main pathogenic macromolecule on the cell surface of gram-negative bacteria such as *Cronobacter sakazakii.* The increase of LPS content in the gastrointestinal tract of ruminants can lead to changes in gastrointestinal epithelial structure and function, and then cause LPS translocation and inflammation [14]. Based on the above-mentioned analyses, we advance two hypotheses: (1) Different farming practices may affect the intestinal microbiota diversity of ruminants; (2) changes in intestinal microbial species can lead to the accumulation of certain metabolites of certain microorganisms, and thus impact the ruminants’ health.

However, since blue sheep and the European mouflon are both endangered animals; almost all of them are kept in zoos, and it is extremely difficult to obtain feces from free-range populations. Therefore, this study selected the feces of the mountain goat (Oreamnos americanus) whose living environment is similar with those of European mouflons and blue sheep for the samples for follow-up research on free-range goats. The mountain goat, a species of Bovidae, is currently distributed throughout the mountains of North America. Phylogenetic analysis based on eight genes showed that blue sheep, the European mouflon, and the mountain goat clustered in the 92% bootstrap confidence level (BCL) value [15]. Therefore, exploring the differences in gut microbes between wild and captive goats in the Xining Wildlife Park could better help rear captive wild animals on the Plateau.

This paper explored the differences in the gut microbial community structures, diversity, and functions between free-range and captive groups of ruminants under different living conditions, aiming to provide some feasible methods for improving the health of captive animals, promoting their adaptation to environmental changes, and preventing disease and parasite outbreaks.

## 2. Materials and Methods

### 2.1. Sample Collection

Five samples of grazing yaks were collected from the Qinghai Lake grazing area (Qinghai Province, China, altitude >2000 m), and four samples of captive yaks were from Datong Hui and Tu Autonomous County (Qinghai Province, China, altitude > 2000 m) from 2 July 2020 to 28 July 2020. All selected adult female yaks (3–4 years old) looked similar in size and received similar immunization procedures and were free of any illness. The dietary data of yaks in the captive and grazing group are given in the Appendix A [16]. The feces in the core of excrement were immediately picked to avoid contamination when the animals defecated, and only the regular excretion was collected for the study. The collected samples were placed in the fecal preservation solution of TIANDZ and then stored in a −80 °C freezer after returning to the Beijing laboratory. The DNA extraction and subsequent tests of the nine samples were simultaneously carried out to rule out human interference in the results. The data on free-range and captive goats were all acquired from the NCBI database. Five samples in the data of free-range goats were acquired from [17] Canada (altitude > 2000 m), with the NCBI sequence read archive (SRA) accession number of PRJNA522005; seven samples of the data of captive goats [18] were acquired from the Qinghai Wildlife Park (Qinghai Province, China, altitude > 2000 m), with the NCBI sequence read archive (SRA) accession number of PRJNA511517. 

### 2.2. DNA Extraction and PCR Amplification 

Microbial community genomic DNA was extracted from fecal samples using an EZNA^®^ Soil DNA Kit [19] (Omega Bio-Tek, Norcross, GA, USA) according to the manufacturer’s instructions. The DNA extract was checked on 1% agarose gel, and the DNA concentration and purity were determined with a NanoDrop 2000 UV-vis spectrophotometer (Thermo Scientific, Wilmington, NC, USA). The hypervariable region V3–V4 of the bacterial 16S rRNA gene was amplified with primer pairs 338F (5′-ACTCCTACGGGAGGCAGCAG-3′) and 806R (5′-GGACTACHVGGGTWTCTAAT-3′) [20] with an ABI GeneAmp^®^ 9700 PCR thermocycler (ABI, Los Angeles, CA, USA). PCR amplification of the 16S rRNA gene was performed as follows [19]: initial denaturation at 95 °C for 3 min, followed by 27 cycles of denaturing at 95 °C for 30 s, annealing at 55 °C for 30 s, extension at 72 °C for 45 s, single extension at 72 °C for 10 min, and end at 4 °C. The PCR mixtures contained 4 µL of 5× TransStart FastPfu buffer, 2 µL of 2.5 mM dNTPs, 0.8 µL of forward primer (5 µM), 0.8 µL of reverse primer (5 µM), 0.4 µL of TransStart FastPfu DNA Polymerase, 10 ng of template DNA, and finally, up to 20 µL of ddH2O. PCR reactions were performed in triplicate. The PCR products were extracted from 2% agarose gel and purified using the AxyPrep DNA Gel Extraction Kit (Axygen Biosciences, Union City, CA, USA) according to the manufacturer’s instructions and quantified using a Quantus™ Fluorometer (Promega, Madison, WI, USA).

### 2.3. Illumina MiSeq Sequencing

Purified amplicons were pooled in equimolar and paired-end sequenced on an Illumina MiSeq PE300 platform/NovaSeq PE250 platform (Illumina, San Diego, CA, USA) according to the standard protocols outlined by the Majorbio Bio-Pharm Technology Co. Ltd. (Shanghai, China). The raw sequencing data from this study were deposited in the genome sequence archive in the BIG data center (https://bigd.big.ac.cn, accessed on 7 May 2022), Beijing Institute of Genomics (BIG), Chinese Academy of Sciences, under the accession number, CRA006829 (7 May 2022).

### 2.4. Processing of Sequencing Data 

The raw 16S rRNA gene sequencing reads were demultiplexed and quality filtered with fastp version 0.20.0 [21] and merged by FLASH version 1.2.7 (Adobe, San Jose, CA, USA) [22] with the following criteria: (i) the 300 bp reads were truncated at any sites with an average quality score of <20 over a 50 bp sliding window, and the truncated reads shorter than 50 bp or containing ambiguous characters were discarded; (ii) only overlapping sequences longer than 10 bp were assembled according to their overlapped sequences. The maximum mismatch ratio of the overlap region was 0.2. Reads that could not be assembled were discarded; (iii) samples were distinguished according to the barcodes and primers, and the sequence direction was adjusted, with exact barcode matching and 2-nucleotide mismatching in primer matching.

Microbiome bioinformatics were performed with QIIME 2 2020.2 [23]. Raw sequence data were demultiplexed and quality filtered using a q2-demux plugin followed by denoising with Divisive Amplicon Denoising Algorithm 2 (DADA2) [24] (via q2-dada2). All amplicon sequence variants (ASVs) were aligned with a multiple alignment program for amino acid or nucleotide sequences (mafft) [25] (via q2-alignment), and used to construct a phylogeny with fasttree2 [26] (via q2-phylogeny). The taxonomy of each ASV representative sequence was analyzed with RDP Classifier version 2.2 against the 16S rRNA database (Silva v138 [27]) using a confidence threshold of 0.7. Rarefaction curves of ASV richness were calculated using a TuTu Analysis Platform.

### 2.5. Difference Analysis between Groups 

Alpha diversity analysis was performed with MicrobiomeAnalyst [28]. Principal co-ordinates analysis (PCoA) visualization and the distance matrix were used to illustrate the correlation between the samples. By coloring the samples with PCoA, selected microbiome-relevant information was described as a clear separation or trend. The correlation network graph was drawn on MicrobiomeAnalyst, using the method of Pearson’s correlation coefficient. Linear discriminant analysis (LDA) and effect size (LEfSe) analyses [29] were used to detect significant differences in the species between the studied groups.

### 2.6. Genome-Wide Functional Prediction 

The whole-genome sequences of *Ruminobacter amylophilus* and *Oscillibacter valericigenes* were downloaded from NCBI and annotated on the KEGG Automatic Annotation Server (KAAS) [30]. KAAS provided the functional annotation of genes by BLAST or GHOST comparisons against the manually curated KEGG GENES database. The result contained KO (KEGG Orthology) assignments and automatically generated KEGG pathways. The annotation results were plotted as heatmaps using the pheatmap package in R.

### 2.7. Correlation Analysis 

Data reflecting the ruminant health status and behavioral activities of yaks and goats from the Plateau with similar living conditions were collected [31,32,33,34,35,36,37,38,39,40,41,42,43,44]. The blood indexes, parasitic infection rates, and grazing behaviors of wild yaks, captive yaks, wild goats, captive blue sheep, and captive European mouflons were analyzed. The correlation analysis was conducted using Spearman’s correlation coefficient with GenesCloud.

## 3. Results

### 3.1. Overview of the Sequencing Data

A total of 1,553,209 high-quality reads were produced from 17 fecal samples (W_Y:5; C_Y:4; W_S:5; C_S:7) and were classified into 2612 ASVs after a quality control of 97% similarity. Rarefaction curves (Appendix A) approached a plateau, which suggested that the number of ASVs was sufficient to reveal the authentic bacterial communities within each sample.

### 3.2. Difference Analysis between Groups

Alpha diversity analysis was performed on yaks and goats with different breeding methods. The abundance-based coverage estimator (ACE) index was used to analyze microbial richness, and the Simpson index was used to analyze microbial diversity. Figure 1 shows that the ACE and Simpson indexes of free-range ruminants were higher than those of the captive, and the ACE indexes of yaks and goats and the Simpson index of goats were statistically significant (*p* < 0.05). Overall, the gut microbial diversity and richness of free-range ruminants were higher than those of captive ruminants. 

Next, Bray–Curtis distance was used for the PCoA analysis of yaks and goats with different breeding methods. The distance between the samples reflected the degree of differences. Figure 2A shows that the contribution rates of the first and second principal components of captive and pasture yaks were 33.6% and 16.6%, respectively, and the two groups of samples had a separation. Figure 2B shows that the contribution rates of the first and the second principal components of the captive and free-range goats were 41.9% and 14.2%, respectively, and there was a separation phenomenon between the two groups of samples. This could be because the composition of gut bacteria of ruminants with different farming methods is significantly different.

We compared and annotated all ASVs with the silva-138-99 database. A Venn diagram and the relative abundance of microbes in the two ruminants are presented in Appendix A at the phylum and family levels, respectively.

The correlation network between the groups was drawn using Pearson’s correlation coefficient at the family level on MicrobiomeAnalyst. Figure 3 shows that the core microbial families of free-range yaks were Oscillospiraceae, Rikenellaceae, and Clostridiaceae, and the core microbial families of yaks in captivity were Ruminococcaceae and Prevotellaceae. The core microbial families of free-range sheep were WCHB1_41, Rikenellaceae, and Lachnospiraceae, and the core microbial families of captive sheep were Spirochaetaceae and Succinivibrionaceae.

LDA was employed to determine the significantly different taxa between the groups from the different breeding methods. Figure 4 shows that the relative abundance of the Succinivibrionaceae family in captive goats and yaks was significantly higher than that in free-range goats and yaks; the relative abundance of the Oscillospiraceae family in free-range goats and yaks was significantly higher than that in captive goats and yaks. This indicated that the bacteria Succinivibrionaceae in the captive ruminants tended to be enriched, while the free-range ruminants tended to have the bacteria Oscillospiraceae enriched. 

### 3.3. Prediction of the Function of Free-Range and Captive Biomarkers 

*Oscillibacter valericigenes,* the most abundant bacteria of Oscillospiraceae, and *Ruminobacter amylophilus,* the most abundant bacteria of Succinivibrionaceae, were used as biomarkers for free-range and captive ruminants, respectively. To explore the special functions of the two bacteria, *Roseburia intestinalis, Clostridium bornimense*, and *Intestinibacter bartlett* from both captive and free-range ruminants were selected as controls. All the five bacteria mentioned above were annotated with KAAS [30], and the abundance of annotated enzymes in each pathway was counted to draw a heatmap. Figure 5 shows that *Oscillibacter valericigenes* was significantly enriched in the enzymes of both the butanoate metabolism pathway and benzoate degradation pathway, while *Ruminobacter amylophilus* was significantly enriched in the enzymes of the lipopolysaccharide-synthesis pathway. Other pathways showed no significant differences.

After constructing the butanoate metabolism and benzoate degradation pathways of *Oscillibacter valericigenes*, it can be seen from Figure 6A that *Oscillibacter valericigenes* had five enzymes in the butyrate metabolism pathway that generated methyl butyrate from acetyl-CoA, among which EC: 2.8.3.8, EC: 4.2.1.17, EC: 1.1. 1.157, and EC: 2.3.1.9 were all annotated, and only EC: 1.3.1.44 and EC: 1.1.1.86 were not annotated. The pathway integrity was 80%, and we speculate that they could cooperate with other bacteria to produce butyric acid. At the same time, six enzymes in the metabolic pathway generated acetyl-CoA from 3-Oxopimeloyl-CoA, among which EC: 2.3.1.16, EC: 7.2.4.5, EC: 4.2.1.17, EC: 1.1.1.157, and EC: 2.3.1.9 were all annotated, and only EC: 1.3.99.32 was not annotated. The pathway integrity was 83%. In the lipopolysaccharide-synthesis pathway of *Ruminobacter amylophilus* (Figure 6B), five enzymes generated ADP-L-glycerol-D-manno-heptopyranose from sedum heptulose-7-phosphate, with EC: 5.3.1.28, EC: 2.7.1.167, EC: 3.1.3.82, EC: 2.7.7.70, and EC: 5.1.3.20. All five enzymes were annotated, and the pathway integrity was 100%. We speculate that they have an extremely strong function in lipopolysaccharide synthesis.

### 3.4. A Correlation Model between Metabolite, Intestinal Microbiota, and Animal Health

To explore the impact of the changes in the gut microbiotas of ruminants on host health under different farming methods, the abundances of enzymes annotated in the pathways mentioned above in captive yaks, free-range yaks, captive goats, and free-range goats were compared. The red blood cell count (RBC), hemoglobin concentration (HGB), hematocrit (HCT), mean corpuscular volume (MCV), parasite infection rate, rumination time, eating time, and breaking and resting time corresponding to different groups (Appendix A) were used for correlation analysis. Figure 7 shows that the parasite infection rate of the host was significantly negatively correlated with benzoate degrading enzymes. Therefore, we speculate that the benzoic acid degradation ability of gut microbes may protect the host from parasitic diseases, while the abundance of LPS synthase was significantly positively correlated with the host rest time. We propose that the stronger LPS synthesis ability of the gut microbes of captive ruminants may be caused by their lack of exercise. In addition, the butyrate metabolism was independent of collected host health factors, which may be related to other aspects of the host.

## 4. Discussion

Several studies have discovered no changes (*p* > 0.05) or even increases in the microbial diversity of captive mammal populations [17,45]. This may be because the samples of the same farming methods were collected from multiple regions, impacting the results, or because the captive culture groups of those studies were kept only for a very short period, and the gut microbes had no time to change. However, most studies have shown that there was a significant decrease (*p* < 0.05) in α diversity in captive ruminant populations, which was consistent with the findings of this study. These diversities may be mainly due to the differences in the food composition of the two groups.

Compared with wild animals, which mainly eat a variety of plants that are high in fiber but low in energy and more difficult to digest, captive animals mainly eat fixed high-energy food. Therefore, free-range ruminants need a more diverse set of gut microbes to help them digest a variety of plants. Theoretically, the high diversity of gut microbial communities is beneficial for the animals to resist external environmental changes [46]. A recent comparative study using 16S rRNA showed significant differences (*p* < 0.05) in the microbiota composition of captive and wild mountain goats [17]. The reason for this difference may be related to the long-term fixed dietary structure of the ruminants mentioned above. Moreover, studies showed that exercise could change the composition of gut microbiotas [47], and it was speculated that another reason for free-range ruminants to have a different gut microbiome composition compared with captive ruminants was because they exercised more every day and needed to find suitable food during exercise.

This study found that the dominant microbiome family of captive ruminants was Succinivibrionaceae, which belongs to the phylum Proteobacteria (Figure 4). Proteobacteria are biomarkers of gut dysbiosis. Dysbiosis occurred because the host’s metabolic disturbances were often accompanied by an increase in Proteobacteria. This might also be the reason why captive animals were more prone to disease. According to the functional annotations of the most abundant bacterium of Succinivibrionaceae, *Ruminobacter amylophilus*, we found that it had a strong potential to synthesize LPS. The correlation analysis showed that this phenomenon was likely significantly related to the low exercise levels of captive animals (*p* < 0.05). Previous studies also found that disturbed gut microbiota could lead to increased gut permeability, resulting in the entry of LPS into the plasma to trigger an immune response [48], while moderate exercise can improve gut permeability [49], and moderate-intensity exercise could limit the increase of intestinal permeability in mice [50]. Combined with the results of this study, we speculate that the shortage in exercise time of captive animals led to a disturbance in their intestinal flora, resulting in increased intestinal permeability; and, because of the enrichment of LPS-producing enzymes in their intestines, their immune responses triggered by LPS were more serious.

The dominant family of bacteria in free-range ruminants is Oscillospiraceae, which belongs to Firmicutes (Figure 4). Firmicutes can help the animal absorb more energy from food [51]. For free-range ruminants, food is in short supply in winter, so they procure energy from foraging to escape predators and maintain body temperature. Their gut microbial structure with a high level of Firmicutes could help them extract as much energy from food as possible to maintain their bodily needs. In addition, by calculating functional annotations on *Oscillibacter valericigenes*, which was the most abundant type of Oscillospiraceae, we found that it had the potential to synthesize butyric acid and degrade benzoic acid, which had a significant positive correlation with ruminants’ resistance to parasitic infections (*p* < 0.05). For herbivorous animals, ingested benzoic acid was combined with glycine to produce hippuric acid within 24 h under the catalysis of enzymes, and then excreted in the urine in the form of hippuric acid (95–100%) [52]. Because benzoic acid and its metabolite, hippuric acid, are both acidic substances, studies have shown that a dietary supplementation of 5000 mg/kg of benzoic acid could reduce the pH of piglets’ digestive chyme, thereby promoting the growth of beneficial bacteria and improving the structure of intestinal microflora [53]. Therefore, combined with the results of this study, we speculate that the enrichment of enzymes related to the degradation of benzoate in the gut of ruminants can improve the utilization rate of benzoic acid and promote the growth of beneficial bacteria to resist parasitic infection. 

However, the understanding of the health effects of both captive and free-range farming methods on ruminants in this study was limited. This study only focused on the effects of gut microbes on host health under different farming methods, and did not consider the impact of breeding methods on the hosts themselves. Moreover, due to the lack of extensive research on the gut microbes of plateau ruminants, the sample size used in this study was not sufficient, and follow-up research should continue to be carried out on other plateau ruminants to update the database. In addition, although the collecting of sample environmental factors was strict in this study, and the data analysis method was relatively rigorous, the speculation that benzoic acid could reduce the infection rate of intestinal parasites in ruminants should also be verified in follow-up studies. 

## 5. Conclusions

In general, by comparing the differences between the two types of ruminants, yaks and goats, under captive and free-range conditions, we found that the composition of gut microbes in captive and free-range ruminants was significantly different. The dominant family shared by captive yaks and goats was Succinivibrionaceae; the dominant family shared by free-range goats and yaks was Oscillospiraceae, *Oscillibacter valericigenes.* This representative species of Oscillospiraceae had the potential to synthesize butyric acid and degrade benzoate, which had a significant positive correlation with the resistance to parasitic infections of ruminants, and *Ruminobacter amylophilus,* a representative bacteria of Succinivibrionaceae, had a strong potential to synthesize lipopolysaccharide. This function was significantly correlated with disease susceptibility in captive animals. Therefore, we concluded that adding an appropriate amount of benzoic acid in the feed and increasing the amount of exercise for the captive animals were beneficial for decreasing their intestinal parasites and improving their health. Taken together, we established a model of the effects of breeding methods on ruminant health (Figure 8).

## Figures and Tables

**Figure 1 life-12-01071-f001:**
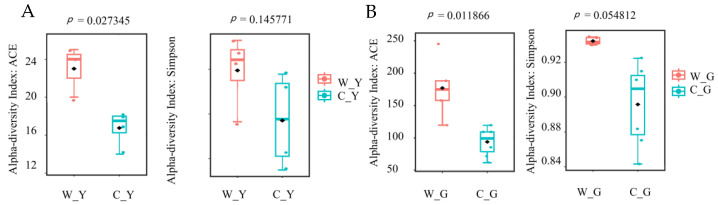
Analysis of gut microbial diversity in captive and free-range ruminants. (**A**) ACE and Simpson indexes of gut microbiota in captive and free-range yaks. (**B**) ACE and Simpson indexes of gut microbiotas in captive and free-range goats. W_Y:Wild Yak; C_Y:Captive Yak; W_G:Wild Goat; C_G:Captive Goat.

**Figure 2 life-12-01071-f002:**
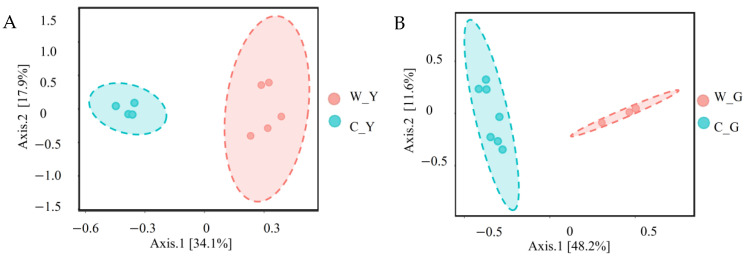
PCoA analysis of gut microbial composition in captive and free-range ruminants. (**A**) PCoA analysis of gut microbes in captive and free-range yaks. (**B**) PCoA analysis of gut microbes in captive and free-range goats. W_Y:Wild Yak; C_Y:Captive Yak; W_G:Wild Goat; C_G:Captive Goat.

**Figure 3 life-12-01071-f003:**
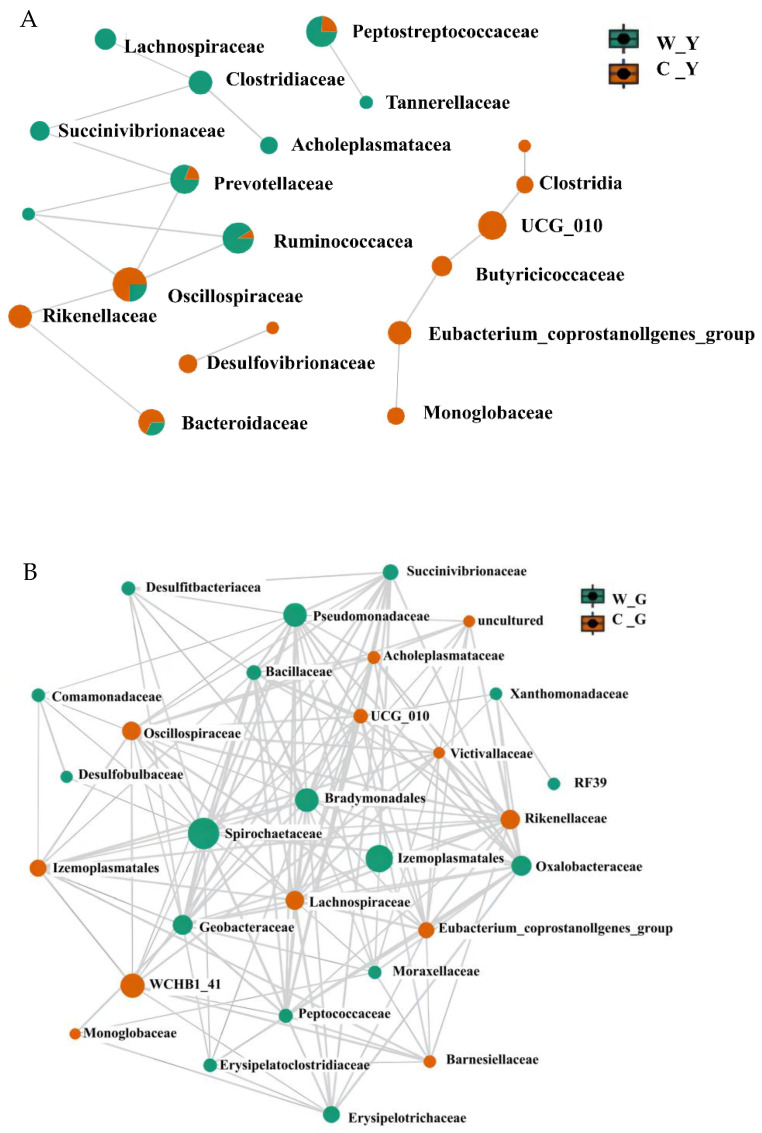
Gut microbial networks in captive and free-range ruminants. (**A**) Interaction network diagram of gut microbes in captive and free-range yaks at the family level. (**B**) Interaction network diagram of gut microbes in captive and free-range goats at the family level. W_Y:Wild Yak; C_Y:Captive Yak; W_G:Wild Goat; C_G:Captive Goat.

**Figure 4 life-12-01071-f004:**
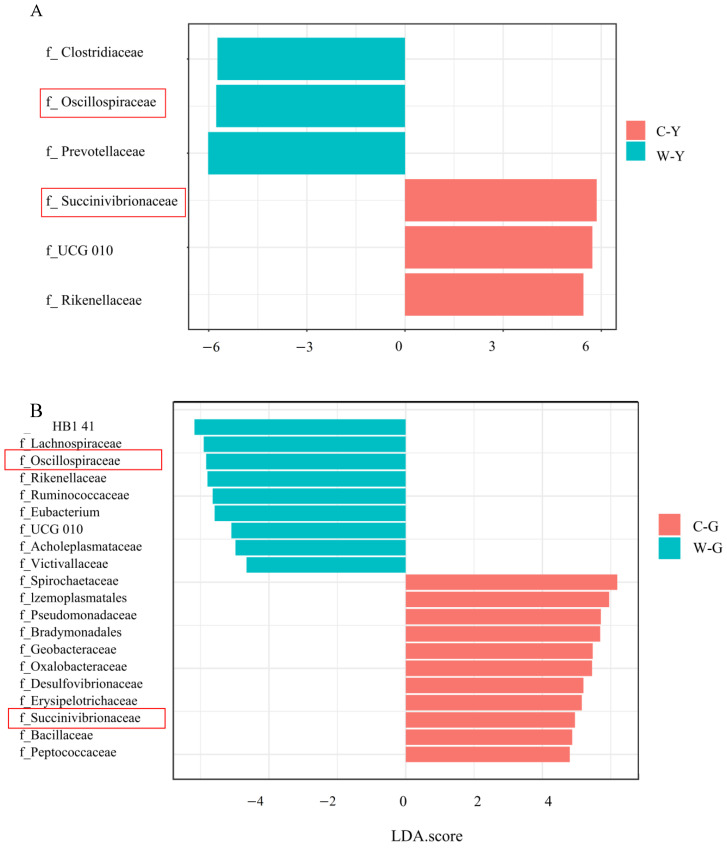
LDA effect size analysis. The histogram shows the biomarkers with statistical differences between groups, and the lengths of the bars indicate the influential degree of the species. (**A**) LDA effect size analysis in captive and free-range yaks at the family level. (**B**) LDA effect size analysis in captive and free-range goats at the family level. W_Y:Wild Yak; C_Y:Captive Yak; W_G:Wild Goat; C_S:Captive Goat.

**Figure 5 life-12-01071-f005:**
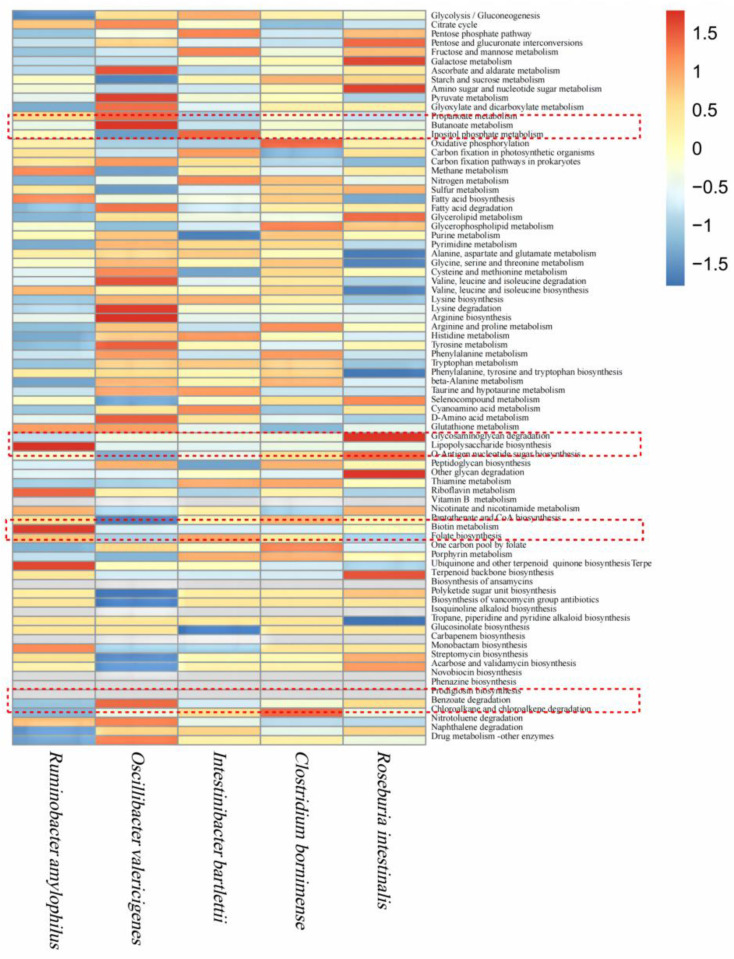
Heatmap of KEGG functional annotation of biomarker and control bacteria in captive and free-range ruminants.

**Figure 6 life-12-01071-f006:**
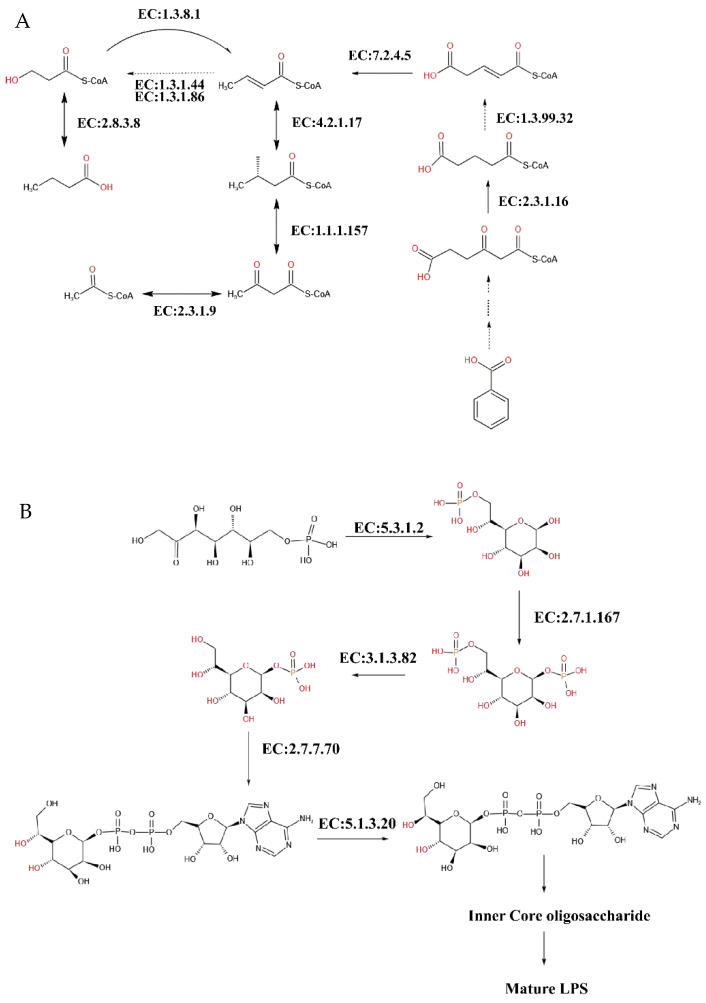
Functional pathway of biomarkers of captive and free-range ruminants. (**A**) Butyrate metabolism and benzoate degradation pathway of *Oscillibacter valericigenes*. (**B**) The lipopolysaccharide metabolic pathway of *Ruminobacter amylophilus*.

**Figure 7 life-12-01071-f007:**
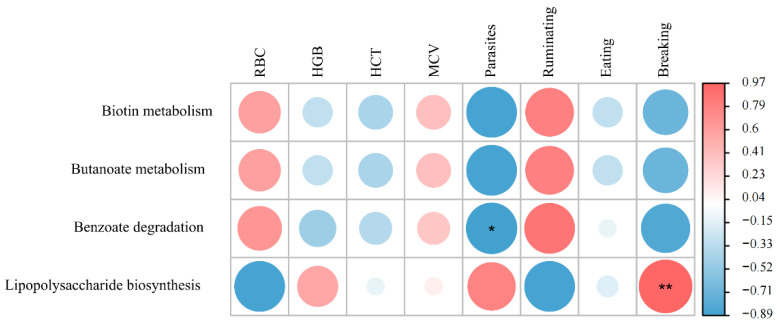
Heatmap of correlation analysis between the abundance of functional enzymes of biomarkers in captive and free-range ruminants and host health indicators. * *p* < 0.05; ** *p* < 0.01.

**Figure 8 life-12-01071-f008:**
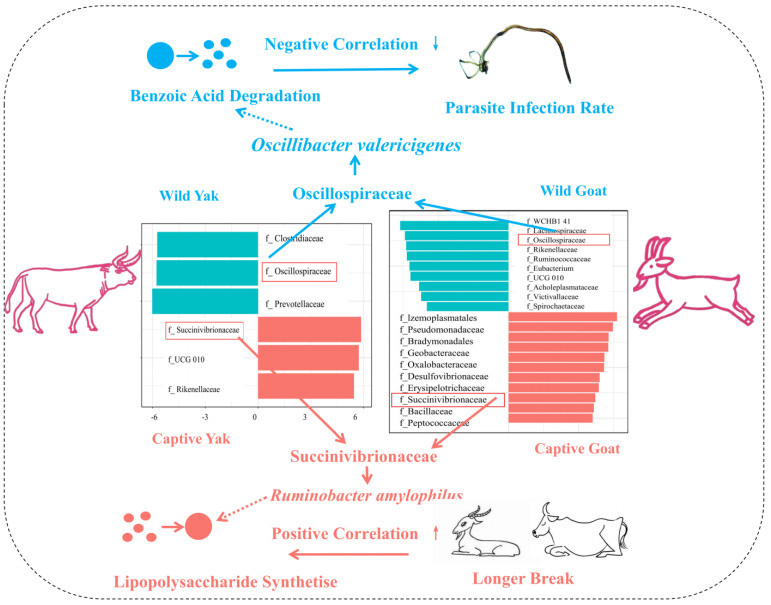
The proposed model; captive ruminants enrich Succinivibrionaceae and wild ruminants enrich Oscillospiraceae, and their potential functions are relative to the health of the animals.

## Data Availability

The raw sequencing data from this study have been deposited in the genome sequence archive in the BIG data center (https://bigd.big.ac.cn, accessed on 7 May 2022), Beijing Institute of Genomics (BIG), Chinese Academy of Sciences, under the accession number: CRA006829.

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
