# Peer review of "Benzoic Acid Metabolism and Lipopolysaccharide Synthesis of Intestinal Microbiome Affects the Health of Ruminants under Free-Range and Captive Mode"

_life, 2022, doi:10.3390/life12071071_

Round 1
Reviewer 1 Report
This manuscript describes the difference in the gut microbiota of two types of ruminants, yaks and sheep, under the conditions of captivity and free-range and studied their effect on benzoate metabolism and lipopolysaccharide synthesis and their correlation with occurrence of parasitic infection and disease susceptibility. This study has due importance and the manuscript can be accepted after minor revision based on the following comments.
1. Change the term Benzoate acid .either use benzoic acid or benzoate in the title and text body of manuscript
2. In line 28 and 29, How current study may be used as a reference for breeding of plateau ruminants
3. The introduction section needs to be improved. Role of benzoic acid and type of lipopolysaccharides must be discussed in it.
4. Material and method= apart from ecological factor , factor related to animals from which sample were taken also effect the gut microbiota such as age, sex, parity, health status etc. so what criteria was chosen for selecting the animals.
5. Describe in detail, sample collection and processing.
6. DNA Extraction and PCR Amplification: add reference of the condition used for amplification in this section.
7. In line 169 and 170 write complete name of abbreviation (DADA2, mafft)
8. In line 176, write the complete name of abbreviation PCoA
9. In line 185 describe the KAAS.
10. In line 188, how ruminant health status and behavioral activities was correlated with current study. As, they vary from place to place
11. In line 200, write the complete name of ACE index
12. In line 257-259 reference is missing.
13. In line 381-382 anti-parasitic infection correct the word.
Reviewer 2 Report
The authors need to address the following raised points below:
Introduction
There are many repetitive sentences across the manuscript and needs to be largely reworked. The introduction section is very lengthy and need to shorten it. The main objective should be coupled with the hypothesis of the study. Study objectives define the specific aims of the study and should be clearly stated in the introduction. The authors clearly mentioned that they selected the feces of the mountain goat (Page 3; line 101) which contradict with statement in the abstract (line 18-19). It is no clear whether the comparison of the differences in the gut microbial community structures, diversity and functions was between yak and sheep OR yak and goat?!
Materials and Methods
The methods information should be adequately provided. The authors did not describe diet/ feeding method of animal kept in captivity? The selection criteria of captive yaks? As the gut microbiota can be dependent on the environment, type of diet given? Additionally, the variation between individual animals should be assessed, or at least taken into consideration. Also, the human handlers can affect the structure of microbiota. These effects must be excluded before considering any significant difference. The method of sample preparation (e.g., equal weight of the fecal sample from each animal being pooled) must be provided. Were the specimen pooled in prior to DNA extraction? The temperature for the storage condition was not mentioned? Did the authors take any precautious for storage such as glycerol storage etc… as we know that the freezing step can undamaged bacteria cells and maybe DNA. I would specify the time of storage, what it the same for all the samples and the condition?
Discussion/Conclusion
The discussion section is well established. However, the statistical analysis is an important part of the work, it need to be clearly summarize and the main results as key messages could be summarize in the conclusion for a better understanding.
Round 2
Reviewer 2 Report
The authors responded adequately to the raised points. The manuscript has been significantly improved and now permits publication in Life.
Author Response
Thanks for your suggestion. We have modified figure 8 to make it easy to read.